# Fluorine Effect for Improving Oxidation Resistance of Ti-45Al-8.5Nb Alloy at 1000 °C

**DOI:** 10.3390/ma15082767

**Published:** 2022-04-09

**Authors:** Wanyuan Gui, Yongfeng Liang, Jingyan Qin, Yongsheng Wang, Junpin Lin

**Affiliations:** 1National Center for Materials Service Safety, University of Science and Technology Beijing, Beijing 100083, China; 2State Key Laboratory of Advanced Metals and Materials, University of Science and Technology Beijing, Beijing 100083, China; liangyf@skl.ustb.edu.cn; 3School of Mechanical Engineering, University of Science and Technology Beijing, Beijing 100083, China; qinjingyanking@foxmail.com; 4College of Materials Science and Engineering, Taiyuan University of Technology, Taiyuan 030024, China; wangyongsheng@tyut.edu.cn

**Keywords:** fluorine effect, surface modification, TiAl alloys, oxidation resistance

## Abstract

In-depth analyses of the anti-oxidation behavior and structure of γ-TiAl alloys are of great significant for their maintenance and repair in engineering applications. In this work, fluorine-treated Ti-45Al-8.5Nb alloys and fluorine-treated oxidized specimens with artificial defects were prepared by isothermal oxidation treatment at 1000 °C. Several characterization methods, including SEM, EDS, XRD and TEM, were used to evaluate the surface microstructure of the fluorine-treated Ti-45Al-8.5Nb alloys and fluorine-treated oxidized specimens with artificial defects. The results indicate that the fluorine promoted the formation of an outer protective film of Al_2_O_3_, which significantly improved the oxidation resistance. The microcracks of oxidized specimens with the artificial defects provided a rapid diffusion passage for Ti and O elements during the 1000 °C/2 h isothermal oxidation treatment process, resulting in the quick growth of TiO_2_ toward the outside. The fine Al_2_O_3_ constituted a continuous film after the 1000 °C/100 h isothermal oxidation treatment. In particular, Al_2_O_3_ particles grew toward the substrate, which was ascribed to the good oxidation resistance and adhesion. These results may provide an approach for the repair of protective oxide film on the surface of blades and turbine disks based on γ-TiAl alloys.

## 1. Introduction

γ-TiAl alloys exhibit broad application prospects in gas turbines and the automotive industry, owing to their low density, high specific strength, high specific stiffness and good high-temperature creep properties [1,2,3,4,5,6,7,8,9]. However, the components of γ-TiAl alloys fail to generate a protective Al_2_O_3_ film at high temperatures (~750 °C) due to the similar formation entropy of TiO_2_ and Al_2_O_3_ [10,11,12,13], resulting in a loose oxide scale with the mixed TiO_2_ and Al_2_O_3_. Therefore, these alloys are facing the challenge of poor oxidation resistance, which has become an obstacle limiting their application.

To tackle this issue, various surface treatments or coatings have been exploited to construct protective oxide films, such as spraying [14,15], plasma surface metallurgy [9,16], electron beam physical vapor deposition (EB-PVD) [17], ion implantation [18] and anodic oxidation [19]. Among them, γ-TiAl alloys treated with halogens including F, Cl, Br and I exhibit strong and long-term protection [18,19,20,21,22,23]. It has been reported that a continuous Al_2_O_3_ scale could be realized at the Al-enriched alloy/scale interface because of the titanium-halide and/or aluminum-fluoride volatilization for γ-TiAl alloys with plasma-based ion implantation. Moreover, a small amount of halogens of TiAl alloy could activate the slow selective Al-oxidation mechanism at the oxide/metal interface. The oxygen ion vacancies in TiO_2_ were cut down due to the halogen ion, leading to the protective Al_2_O_3_ scale and sluggish growth of the TiO_2_ scale on TiAl alloys. Recently, the fluorine ion was introduced into high Nb-TiAl alloys by the anodizing oxidation method. Despite the presence of etching and spotty defects on the surface, the high temperature oxidation resistance of the alloy could still be improved.

In view of engineering applications, the convenient and inexpensive methods to introduce fluorine into large-sized γ-TiAl components have received extensive attention, such as dipping and spraying. For instance, the NH_4_F solution-impregnated γ-TiAl alloy could form a dense and continuous Al_2_O_3_ film and thus improve the anti-oxidation [22]. The surface of γ-TiAl alloys were immersed into the HF solution for long-term protection [24]. In order to meet the low-cost demands of industrial applications, the smart reintroduction of fluorine in the fluorine-treated γ-TiAl alloys system offers a highly attractive solution for engineering repairs, particularly in repairing surface protective oxide film on blades and turbine disks based on γ-TiAl alloys. 

In this work, the fluorine effect was applied for improving the high temperature oxidation resistance of the Ti-45Al-8.5Nb alloy and its oxides. The fluorine-treated γ-TiAl alloy and its oxidized specimen with artificial defects were investigated by isothermal oxidation at 1000 °C. Furthermore, SEM and TEM characterizations revealed the evolution of the phase and microstructure of the oxidized specimens with artificial defects.

## 2. Experimental Section

### 2.1. Material and Treatment

A casting γ-TiAl alloy ingot with a nominal composition of Ti-45Al-8.5Nb (at %) alloys was fabricated by a vacuum induction melting furnace. The ingot was cut into the block-shape specimen (10 × 10 × 5 mm^3^), and the oxidation tests were carried out by wire electrical discharge machining. These specimens were ground and polished according to the metallographic specimen preparation process. The NaF-containing aqueous solution (0.15 mol/L) was sprayed onto the surface of γ-TiAl alloy specimens, then washed with deionized water and dried.

### 2.2. Oxidation Treatment

Oxidation experiments were carried out at 1000 °C using a tube furnace. These prepared specimens were placed in quartz tubes to ensure the loss of oxide flakes, and the total mass of quartz tube was weighted every 10 h with an analytical balance. After isothermal oxidation for 100 h at 1000 °C using a Rockwell hardness tester (HRC, 150 kg), the artificial microcracks were introduced onto the surface of specimens treated with the NaF aqueous solution. Then, these specimens were sprayed again with the NaF-containing aqueous solution (0.15 mol/L), and the isothermal oxidation was repeated for 2 h and 100 h, as shown in Figure 1. 

### 2.3. Characterization

The phase formation and crystal structure of the samples were determined by X-ray diffraction (XRD, Rigaku DX-2700, Tokyo, Japan). The surface and cross-section morphologies of the samples were observed by field emission scanning electron microscopy (FESEM, ZEISS Gemini 300, Oberkochen, Germany) at an acceleration voltage of 15 kV, and the composition was detected via energy dispersion spectrometry (EDS) with a take-off angle of 36.5° and live time of 2 min to further improve the quantitative accuracy. Prior to the SEM/EDS tests, the samples were washed with ethanol and dried in a vacuum oven (60 °C). Moreover, the microstructure and chemical composition were examined by the transmission electron microscope (TEM, JEOL JEM-2100, Japan) and selected area electron diffraction (SAED) patterns. The wetting behavior of the NaF-containing aqueous solution (0.15 mol/L) droplets on γ-TiAl was investigated. The volume of each droplet was approximately 5 μL, and both the side and top view of each droplet were recorded during the contact angle measurements.

## 3. Results and Discussion

### 3.1. Microstructure

Figure 2 shows the microstructure of the TiAl alloy. The SEM image revealed that the alloy was composed of γ-TiAl phase and Ti_3_Al phase (α_2_) in the form of a near-lamellar structure.

### 3.2. Oxidation Kinetics

Since the wettability of the NaF-containing aqueous solution drops on γ-TiAl alloy exerted an important influence on the oxidation process, a contact angle test was performed to evaluate the effect of the NaF solution droplets on the wetting performance on the sample surface. As shown in Figure 3, the initial surface of the pristine Ti-45Al-8.5Nb alloy sample was hydrophilic with a contact angle of 39.6° (Figure 3a). The results indicate the excellent wettability of the interface between the NaF-containing aqueous solutions and Ti-45Al-8.5Nb alloys. In addition, Figure 3b,c display the contact angle of the fluorine-treated γ-TiAl alloy and bare TiAl alloy after isothermal oxidation. The results suggested that the contact angle increased after 1000 °C/100 h isotherm oxidized treatment, and the fluorine treatment was beneficial to improve the wettability of the oxidation on the Ti-45Al-8.5Nb alloy’s surface.

Figure 4 shows the oxidation kinetics of isothermal oxidation of the γ-TiAl alloy. The mass gain curves increased linearly to 6.03 mg/cm^2^ for the bare Ti-45Al-8.5Nb alloys, while those of the fluorine-treated Ti-45Al-8.5Nb alloy specimens increased to 0.81 mg/cm^2^. These results indicate that the oxidation resistance of the Ti-45Al-8.5Nb alloys was improved by the NaF-containing aqueous solution treatment, in agreement with the previous results [19,22,25]. The pale yellow oxide scale in a curled state, as shown in the inset, had peeled off the surface of the specimen. In contrast, the fluorine-treated specimen had a deep yellow oxide scale on the surface without any exfoliation.

### 3.3. Structure and Chemical Composition

Figure 5 shows the XRD analysis of the Ti-45Al-8.5Nb alloy specimens after isothermal oxidation with and without fluorine treatment. Oxides (such as TiO_2_ and Al_2_O_3_) and intermetallics (such as γ and α_2_) were found in the spectra of both specimens. Combined with the images in Figure 3, the oxide scale on the bare TiAl specimen was separated from the surface during oxidation, which was identified to be the mixture of TiO_2_ and Al_2_O_3_. Similar observations were reported in the study of Zhao et al. [26], in which TiO_2_ and Al_2_O_3_ exhibited competitive growth due to their similar formation energies, leading to the failure protection for the substrate alloy [6,8,26,27]. However, the oxidized TiAl specimen with fluorine treatment presented an increased diffraction peak intensity of α_2_ and increased oxides, such as the crystal plane of (201) at 40.8° for α_2_, and crystal plane of (113) at 43.26° for α-Al_2_O_3_. 

The surface and cross-sectional morphologies of the fluorine-treated specimen after isothermal oxidation are shown in Figure 6. The oxide scale displayed a smooth surface, despite several bulges (Figure 6a). The fine oxides with a size of ~180 nm to 1 μm, mainly composed of Al_2_O_3_, were observed on the smooth region under a high magnification, as marked by point 1 in Figure 6b and Table 1. However, according to EDS results, the oxide particles with a dimension of ~1.3–4.2 μm were identified as TiO_2_ of the bulges, as marked by point 2 in Figure 6c, which was also detected with a rutile character on the top layer of the cross-section (Figure 6d). Moreover, a hierarchical structure including the Al_2_O_3_-rich top layer, Al-depleted subsurface and Nb-rich layer was found based on the EDS mapping analysis (Figure 6e,f). Due to the presence of fluorine, the Al_2_O_3_ was preferentially generated and ascribed to the outer oxide film [19,20,24]. Therefore, the subsurface had an Al-depleted but Ti-rich layer. Such oxidation and diffusion toward the outside resulted in a discontinuous layer with higher Nb content than the substrate [19,21], as marked by point 10 in Figure 6e. Owing to the strong outside diffusion of Ti and Al elements, the formation of TiO_2_ and Al_2_O_3_ led to the depletion of Ti and Al elements, which promoted the enrichment of Nb at the oxide scale/TiAl substrate interface. The Nb-rich phase was identified as AlNb_2_ [28,29], and has a positive role in limiting the inward diffusion of the O element [29,30]. Similar results were reported by Schutze et al. [31,32,33], in which the increase in the fluorine effect by adding Nb as a corridor could selectively transit the Al rather than Ti element to form Al_2_O_3_, according to the thermodynamic calculation of the involved MeF (Me = Ti, Al) and MeO phases at different oxygen partial pressures.

Repairing the components with macro- and/or micro defects plays a vital role in practical applications, such as in blades and turbine disks. Firstly, the fluorine-treated γ-TiAl specimens were oxidized for 100 h, scratched with an indenter and then treated again with the NaF solution, followed by a second oxidization. The oxidized specimens with fluorine treatment were subjected to a second treatment and oxidation to elucidate the formation mechanism of oxides from the oxide scale with microcracks. Figure 7a,e show the impressions of the specimens after oxidation for 2 h and 100 h at 1000 °C. It can be seen that the crowded oxide particles protruded from the microcracks, generating some visible strips (Figure 7b,f). The EDS mapping scan results (Figure 7d,g) indicate that these oxide strips were filled with TiO_2_ particles after 2 h of oxidation (point 11), and with the Al_2_O_3_ and TiO_2_ mixture of the specimen after 100 h of oxidation (point 14). The chemical compositions of unbroken regions (points 12 and 13) were consistent with the results of isothermal oxidation-treated TiAl specimens. Several chemical reactions existed among the F element and these Al, Ti and Nb metals.

For the current second-treated, oxidized TiAl specimens, the cracked surface was contacted firstly with the F solution. Compared with other fluorides [34,35], the Al monofluoride on the surface of specimens displayed the highest vapor pressure. Because of the vapor vocalization of the Al monofluoride, the residual Ti element on the surface of specimens was re-oxidized to TiO_2_ at a high temperature, resulting in the visible strips of TiO_2_-rich particles. At the oxide scale, the oxygen partial pressure dropped steeply. The Al monofluoride could react with the oxygen to generate Al_2_O_3_ at a high temperature (900 °C) with the lowest formation energy [32,33,34]. Therefore, the large-sized Al_2_O_3_ particles were embodied into the phases of the substrate.

### 3.4. Microstructure and Interface

In order to explore the microstructure, TEM and HRTEM were utilized to characterize the fluorine-treated specimen with the artificial defect after oxidation at 1000 °C for 100 h. The specimen was sliced by the FIB near the artificial defect. An oxide scale with several voids had a thickness of ~1.7 μm according to the completed specimen (Figure 8a), and consisted of the mixed Al_2_O_3_ and TiO_2_ top layer. Beneath this layer, there was a compact and continuous Al_2_O_3_ film with a thickness of 500 nm. Furthermore, the distribution between the Al/O and Ti/Nb elements showed a complementary relationship, based on EDS mapping of Ti, Al, Nb and O elements (Figure 8b). The results suggest the formation of the Al_2_O_3_ particles with a size of 1–5 μm. The Nb element has an ionic radius of 0.64 Å (Nb^5+^), similar to the Ti^4+^ of 0.605 Å [36,37]. In addition, as a stable element for β-Ti alloys, the Nb element exhibited an ultimate mutual solubility [38], but a doped effect on the oxides, especially occupying the Ti sites in TiO_2_ to form the possible phases, such as (TiNb)O_2_ and Al_2_(TiNb)O_5_ [28,29,30]. These phases retarded the outside diffusion of the Ti element. The TiO_2_ particle in the outermost layer was observed by the SAED pattern (Figure 8c, (c1)). The continuous Al_2_O_3_ film was identified as the α phase (Figure 8d, (d1)). Combined with the XRD result of thermal oxidation, it can be inferred that the protective effect is generated by this continuous α-Al_2_O_3_ film. The typical α_2_-Ti_3_Al phase can also be identified in Figure 8e, (e1)). Several large Al_2_O_3_ particles with a size of ~1–5 μm grew among γ-TiAl and α_2_-Ti_3_Al phases under the oxide scale (Figure 8a,e). Moreover, the α_2_-Ti_3_Al and Al_2_O_3_ share a tight interface under the oxide scale (Figure 8e,f). The growth pattern of these Al_2_O_3_ particles embodied into phases of the substrate could enhance the adhesion of oxide scale/alloy. Notably, previous work reported similar results [31,32,33,39], in which the generation of Ti-fluoride and Al-fluoride might arise from the chemical reactions between the fluorine and metals (including Ti and Al) by fluorine treatment, such as ion implantation, anodic oxidation and spray. After a short oxidation time, exclusive Al_2_O_3_ and minor TiO_2_ were formed on the surface of the specimens due to the volatilization of Ti-fluoride. With the prolongation of oxidation, the fluorine would promote the formation of a continuous and dense Al_2_O_3_ layer, consistent with the chemical element distribution shown in Figure 5 and Figure 7. The oxidation approaches of the relevant reactions followed Me (Me = Ti, Al) → MeF (fluorides) → MeOF (oxy-fluorides) → MeO (oxides); or/and Me → MeF → MeO [34]. The continuous Al_2_O_3_ layer hindered the internal diffusion of oxygen and the external diffusion of metals. Under the Al_2_O_3_ layer, the fluorine-aided selective transport of Al by the Al-fluoride was related to the partial pressures of p^(AlF)^ and p^(F2)^ [31,40], e.g., the minimum p^(F)^ of Ti_0.5_Al_0.5_ was 1.30 × 10^−8^ bar at 1000 °C. If the balance at the local region of the specimen was disrupted or exfoliated due to the failure of the oxide scale during the oxidation process [26], Ti and Al would diffuse competitively from the damaged region, followed by an accelerated failure of the oxide scale. The current experimental procedure followed the order of fluorine-treated TiAl alloy → isothermal oxidation → cracked specimen → fluorine treatment again → isothermal oxidation again. For the current second-treated, oxidized TiAl specimens, the cracked surface was firstly contacted with the F solution. The Al monofluoride on the surface of specimens displayed the highest vapor pressure of all other fluorides [34,35]. Due to the vapor vocalization of the Al monofluoride, the residual Ti element on the surface of specimens was re-oxidized to form TiO_2_ during the second oxidation at a high temperature, leading to the visible strips of TiO_2_-rich particles. Under the oxide scale, the oxygen partial pressure decreased steeply. The Al monofluoride reacted with the oxygen to generate Al_2_O_3_ at a high temperature (900 °C) with the lowest formation energy [32,33,34]. The artificial microcracks performed as rapid diffusion passages for Ti, Al and O elements. The activity and diffusion rates of Ti were higher than those of Al, leading to a quick growth of TiO_2_ toward the outside in a short time. However, the fine Al_2_O_3_ formed a continuous film at the bottom of the oxide scale after isothermal oxidation. In particular, Al_2_O_3_ particles instead of TiO_2_ particles grew toward the substrate and embodied in the γ-TiAl and α_2_-Ti_3_Al phases, resulting in their high adhesion. Therefore, the γ-TiAl alloys treated by fluorine exhibited excellent anti-oxidation properties at a high temperature.

## 4. Conclusions

In this study, we executed a simple and effective strategy involving a two-step spraying and oxidation process to produce Ti-45Al-8.5Nb alloys with a dense Al_2_O_3_ film for application at high temperatures. The main results are as follows:
(1)The oxidation resistance was significantly improved due to the formation of a protective Al_2_O_3_ film induced by the fluorine effect during isothermal oxidation at 1000 °C (6.03 mg/cm^2^ weight gain for the bare Ti-45Al-8.5Nb alloys, but 0.81 mg/cm^2^ for the fluorine-treated Ti-45Al-8.5Nb alloys).(2)The Ti diffused along the artificial microcracks, resulting in a quick growth of TiO_2_ in a short time (2 h), while a continuous Al_2_O_3_ film was generated underneath the oxide scale during further isothermal oxidation (100 h).(3)The Al_2_O_3_ particles between the alloy and continuous Al_2_O_3_ film were reflected in the γ-TiAl and α_2_-Ti_3_Al phases, which improved the adhesion of the oxide scale (fluorine-treated specimen had a deep yellow oxide scale on the surface without any exfoliation). These findings represent significant progress towards reducing the high temperature oxidation resistance of TiAl intermetallics by surface fluorine modification.


## Figures and Tables

**Figure 1 materials-15-02767-f001:**
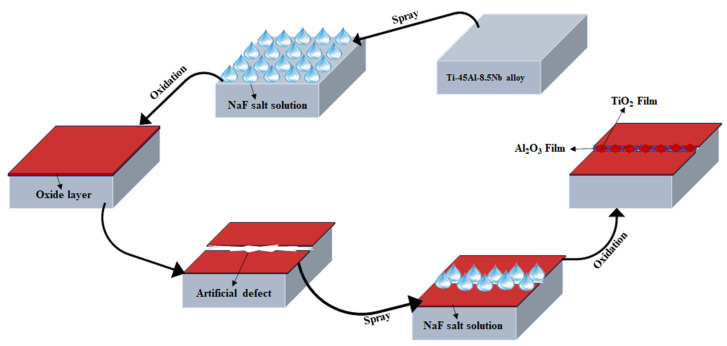
X The diagram of the γ-TiAl alloy specimens treated with NaF-containing aqueous solution.

**Figure 2 materials-15-02767-f002:**
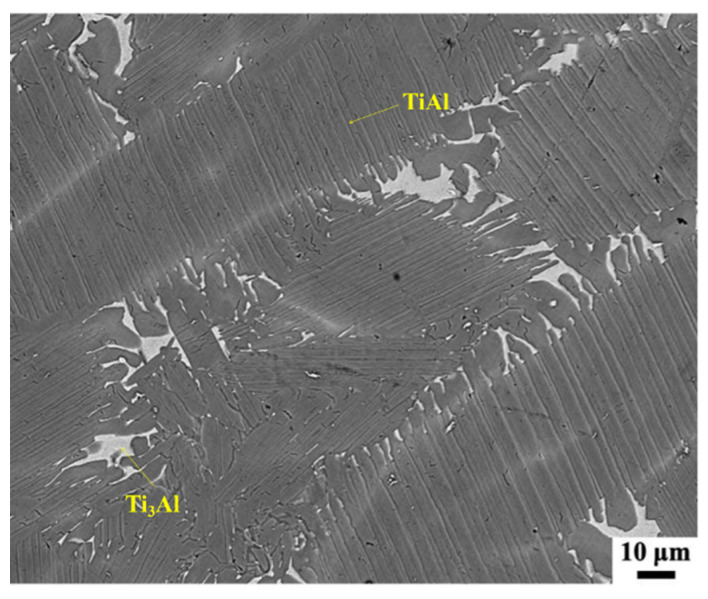
SEM image of the TiAl alloy.

**Figure 3 materials-15-02767-f003:**
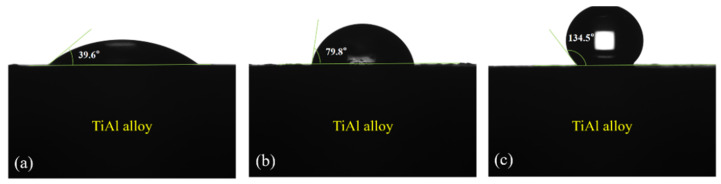
The wettability of the NaF-containing aqueous solution on the γ-TiAl alloys: (**a**) pristine γ-TiAl alloy, (**b**) fluorine-treated γ-TiAl alloy after isothermal oxidation, (**c**) bare TiAl alloy after isothermal oxidation.

**Figure 4 materials-15-02767-f004:**
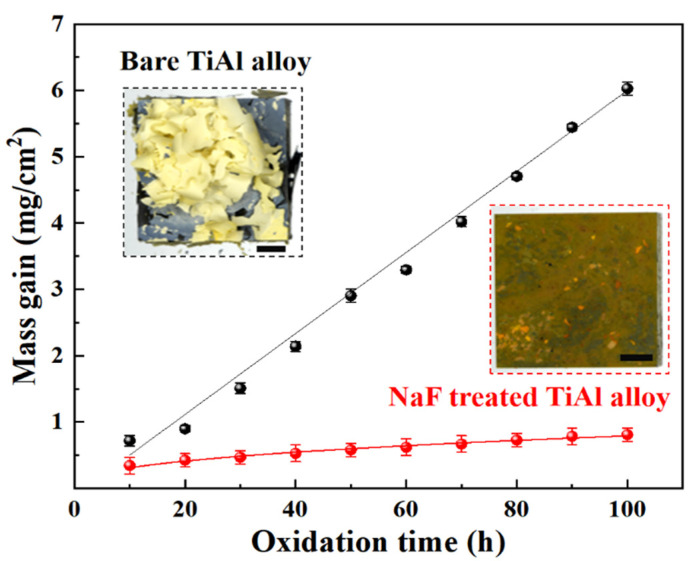
The mass gain curves of Ti-45Al-8.5Nb alloy specimens with and without fluorine treatment during isothermal oxidation at 1000 °C.

**Figure 5 materials-15-02767-f005:**
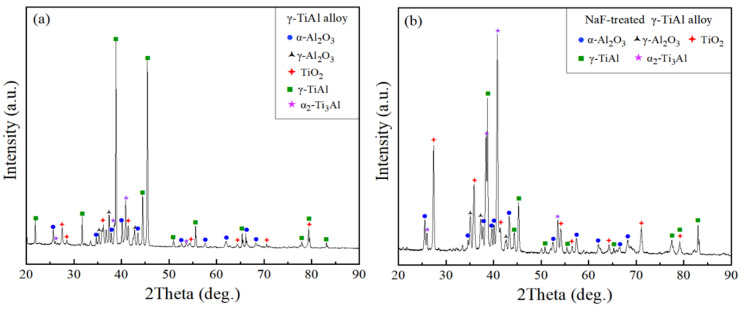
XRD diffraction spectra of the Ti-45Al-8.5Nb alloys after 1000 °C/100 h isothermal oxidized treatment (**a**) without and (**b**) with fluorine spray.

**Figure 6 materials-15-02767-f006:**
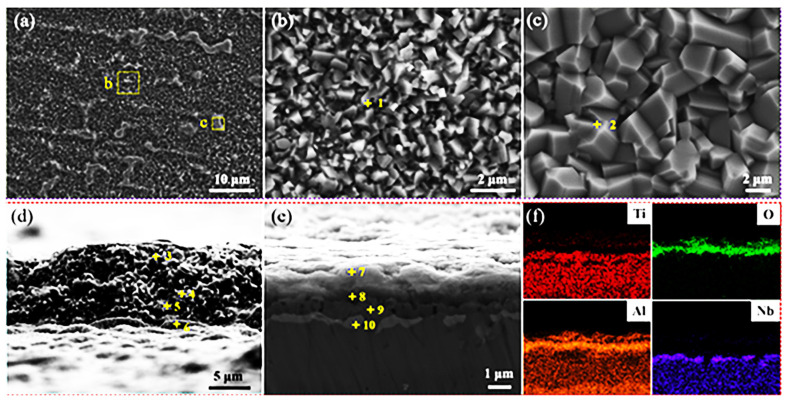
(**a**) Surface morphology of the fluorine-treated specimen after oxidation at 1000 °C for 100 h, (**b**,**c**) the local regions at high magnification, (**d**) SEM side-view of the oxide scale top layer, and (**e**) the cross-section morphology and (**f**) its EDS mapping results.

**Figure 7 materials-15-02767-f007:**
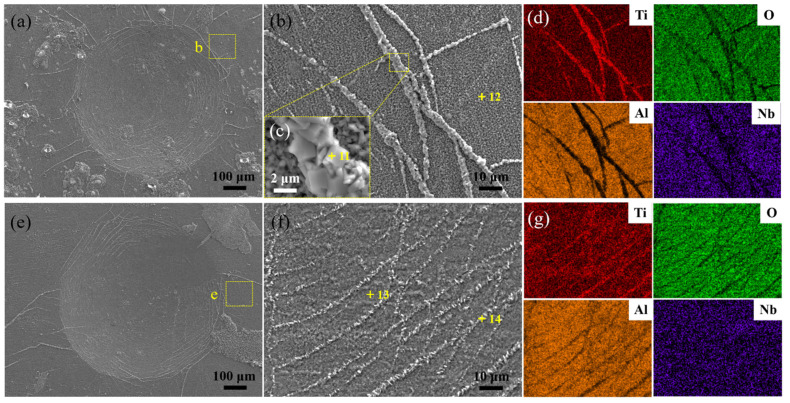
(**a**,**e**) The impressions of specimens after the oxidation for 2 h and 100 h at 1000 °C, (**b**,**f**) their local SEM images, (**c**) the inset marked by yellow square frame in (b) and (**d**,**g**) the corresponding EDS mapping results.

**Figure 8 materials-15-02767-f008:**
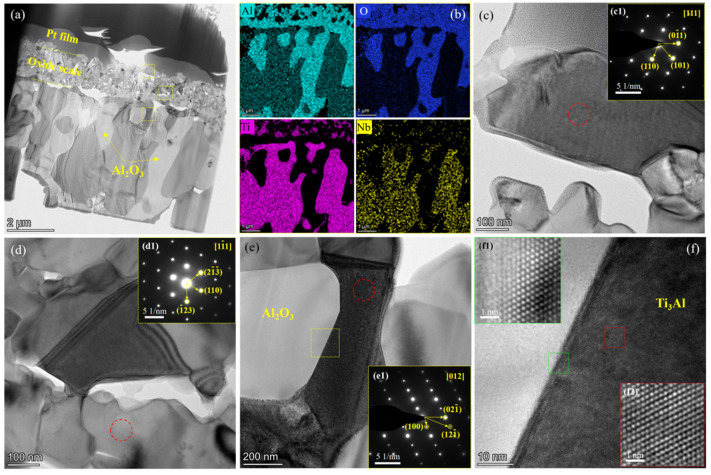
(**a**) TEM image of oxide scale which was selected near the defect of the fluorine-treated specimen with the artificial defect after oxidation at 1000 °C for 100 h, (**b**) the corresponding chemical elements distribution of STEM images. (**c**–**f**) The marked local regions in (**a**) by yellow square examined at high magnification. The red circle region examined by SAED pattern including (**c1**–**f1**). The yellow square region in (**e**) investigated by HRTEM in (**f**). The green square (f1) and red square regions (f2) in (**f**) examined by inverse FFT pattern.

**Table 1 materials-15-02767-t001:** EDS results derived from the marked points in Figure 6 and Figure 7.

Point	Composition/at %			
	Ti	Al	Nb	O
1	11.07	20.30	0.75	67.86
2	25.19	0.46	0.67	73.67
3	19.93	2.25	0.38	77.44
4	14.27	7.66	0.49	77.58
5	12.42	9.72	0.34	77.52
6	9.84	13.87	0.31	75.98
7	9.91	24.26	0.98	64.85
8	2.64	36.84	0.26	60.26
9	54.44	18.34	1.48	25.75
10	45.11	27.78	14.77	12.34
11	25.21	1.16	0.02	73.61
12	6.33	25.42	0.90	67.36
13	5.02	26.64	0.47	67.87
14	11.52	11.70	0.27	76.50

## Data Availability

The data that support the findings of this study are available from the first corresponding author upon reasonable request.

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
