# Peer review of "Fluorine Effect for Improving Oxidation Resistance of Ti-45Al-8.5Nb Alloy at 1000 °C"

_materials, 2022, doi:10.3390/ma15082767_

Round 1

Reviewer 1 Report

The manuscript entitled “Fluorine effect for improving oxidation resistance of Ti-45Al-2 8.5Nb alloy at 1000 ℃” has been submitted to be considered for publication on Materials. The authors presented the spraying method to treat the pristine as well as dented surface of the alloy before oxidizing it and characterized the oxidized the sample to reveal the fluorine effect on enhanced oxidation resistance. The thema and results are interested to readers of Materials. However, further improvement needed to justify the further consideration on publication. Please consider the following comments:

  1. On the sample preparation and the method of fluorinate the surface. Since the authors used spraying as method of choice, followed by cleaning the surface and heated it. It is necessary to characterize the surface roughness, as well as wetting behaviour of the NaF solution on to the surface. Please discuss on how the NaF interacts with the pristine alloy surface. Does varying the surface roughness, or concentration of NaF result on the change of oxidation resistance of treated samples ? How does this spraying different from dipping the same sample in the same solution, followed by cleaning and heat treated?
  2. The authors created the “defect” by indentation of the surface. This will not remove the previous treated surface. What will happen if instead of indentation, scratching is applied?
  3. Page 3 and 4, the authors discussed about the oxidation of the treated and pristine sample and the oxidation products that formed on the surface using XRD and EDS. The pristine sample produced a TiO2 and Al2O3 as oxidation products, as stated in line 102. The authors wrote “However, the diffraction peaks intensity of α2 and oxides increased for the 105 oxidized TiAl specimen with fluorine treatment such as the crystal plane of (201) at 40.8° for α2, and crystal plane of (113) at 43.26° for α-Al2O3. It indicated the preferential growth of these phases, which related to the improved oxidation resistance.”  However, the XRD data showed both TiO2 peaks and Al2O3 and the TiO2 peaks are stronger. How can they come to conclusion of “preferential growth of these phases” (Al2O3) ?  
  4. Figure 5 shows the EDS results on a not flat sample. Thus the result from point analysis 5d is not representative. The mapping is performed on a very small area, how can this data be presentative for the whole sample? For example, in the 5f a Nb enrichment can be seen, but this is not the case in Fig 7b.
  5. How precise is the EDS measurement? How can the authors make sure that the EDS data has statistical meaning? Normally, EDS is performed without standard samples, and thus only qualitative.
  6. The authors claim the microcracks on the indented sample without clear image of the crack, or cross-section that prove that there are really cracking after indenting. Please provide at least high magnification of the cracks. In my opinion, the lines in 6b and 6f can not be related to the microcracks. If so, these cracks are quite strange. Figure 6 is also very difficult to understand.
  7. To confirm the effect of fluorine treated sample, the same SEM, TEM analysis should be presented on the non-treated sample. Without this control information, one can not draw a conclusive statement on the effect of treated samples. 

Author Response

Reviewer #1 (Comments to the Author):

  1. On the sample preparation and the method of fluorinate the surface. Since the authors used spraying as method of choice, followed by cleaning the surface and heated it. It is necessary to characterize the surface roughness, as well as wetting behaviour of the NaF solution on to the surface. Please discuss on how the NaF interacts with the pristine alloy surface. Does varying the surface roughness, or concentration of NaF result on the change of oxidation resistance of treated samples? How does this spraying different from dipping the same sample in the same solution, followed by cleaning and heat treated?

Response: Good questions! The NaF salt solution (0.15 mol/L) was sprayed onto the surface of γ-TiAl specimens. It is different with the HF acid. The HF acid could corrode the surface of γ-TiAl specimens. The NaF salt solution (0.15 mol/L) cannot change the surface of γ-TiAl specimen, although we can make up this experiment to observe the surface roughness of the-treated specimens. The NaF interacts with the pristine alloy surface, which has been modified as “It has been reported that the F negative ion could react with the Ti and Al to form fluorides (such as TiF3 and AlF3) [7-12].” in the Figure 1. The NaF solution has the perfect wetting on the surface of γ-TiAl specimen. If the specimens with the different original surface roughness (such as polishing by different sandpaper 600#, 1000#, 2000#, 3000# and 5000#) were treated by the current sprayed -NaF salt solution, we infer that the surface roughness will change with the original surface roughness of the specimens. We will conduct further experimental on the basis of this idea. Because of the NaF solution has the perfect wetting on the surface of γ-TiAl specimen, the spraying method has the same effect with the dipping method.

  1. The authors created the “defect” by indentation of the surface. This will not remove the previous treated surface. What will happen if instead of indentation, scratching is applied?

Response: Great, that is a good idea. I think if instead of indentation the scratching is applied, the surface will show interesting morphology which may consisted with Al2O3 and TiO2 on the pristine alloy surface. We will conduct more experimental about this idea in next work.

  1. Page 3 and 4, the authors discussed about the oxidation of the treated and pristine sample and the oxidation products that formed on the surface using XRD and EDS. The pristine sample produced a TiO2 and Al2O3 as oxidation products, as stated in line 102. The authors wrote “However, the diffraction peaks intensity of α2 and oxides increased for the 105 oxidized TiAl specimen with fluorine treatment such as the crystal plane of (201) at 40.8° for α2, and crystal plane of (113) at 43.26° for α-Al2O3. It indicated the preferential growth of these phases, which related to the improved oxidation resistance.” However, the XRD data showed both TiO2 peaks and Al2O3 and the TiO2 peaks are stronger. How can they come to conclusion of “preferential growth of these phases” (Al2O3)?

Response: Good suggestion! We had deleted the sentence “It indicated the preferential growth of these phases, which related to the improved oxidation resistance.”. The direct evidence of the improved oxidation resistance for NaF solution treated γ-TiAl specimens was shown in Fig. 3.

Fig. 3 The mass gain curves of γ-TiAl alloy specimens with and without fluorine treatment during isothermal oxidation at 1000℃.

  1. Figure 5 shows the EDS results on a not flat sample. Thus the result from point analysis 5d is not representative. The mapping is performed on a very small area, how can this data be presentative for the whole sample? For example, in the 5f a Nb enrichment can be seen, but this is not the case in Fig 7b.

Response: In figure 5, we used (a) Surface morphology of the fluorine-treated specimen after oxidized at 1000℃ for 100 h, (b) and (c) the local regions at high magnification. It is the surface morphology, and chemical composition by EDS points analysis including “1 (smooth surface) and 2 (bulges)”. The characters are different at regions of (b) smooth surface and (c) bulges, which is the representative characterization of Fig. 5 (a). Therefore, we utilized the Fig. 5(d) to study the bulges from SEM side-image of the oxide scale top layer. Of course, the chemical composition by EDS points analysis including “3, 4, 5 and 6” were studied from the oxide scale top layer to the sub-surface. Then, Fig. 5(e) shows the cross-section morphology under the smooth surface and (f) its EDS mapping results. During oxidation at high temperature, the oxides including Al2O3 and TiO2 grew through the outward diffusion of Ti and Al elements but inward diffusion of Oxygen. As a result, the phase and micro structure would change. So, we employed the TEM to study the phases and their relationship nearby surface and interface in Fig. 7. Some case between Fig. 5f and Fig 7b is that oxides scale at outside consisted of the Al2O3 and TiO2, but the Nb-enrich phase (Fig. 7b) or layer (Fig. 5f) exited under the oxides scale.

  1. How precise is the EDS measurement? How can the authors make sure that the EDS data has statistical meaning? Normally, EDS is performed without standard samples, and thus only qualitative.

Response: Yes! Normally, EDS is performed without standard samples, and thus only qualitative. Here, the EDS point scanning was utilized to test the basic chemical compositions in table 1. We appreciate your beneficial suggestions, and understand your concerns for quantitative analysis of EDS measurements. The descriptions about the sample preparations and important parameters for SEM observations and EDS accuracy have been supplemented in the manuscript. In the experiments, the samples used for observing surface and cross-section microtopography were washed with ethanol and dried in a vacuum oven (60°C) until SEM/EDS tests. Afterwards, the whole SEM/EDS examinations for the samples were performed in high vacuum (System Vacuum: 2 x 10-5 mbar), and thus the oxygen self-absorption on samples should be negligible. In addition, the accuracy of EDS method for light elements should indeed be considered. Moreover, a relatively low acceleration voltage of 10.0 kV, take-off angle of 36.5 deg, and live time of 2 min have been used for further enhancing the EDS precision for quantification. Based on the EDS results, we can roughly infer the main phase or oxide-enrich according to these results.

  1. The authors claim the microcracks on the indented sample without clear image of the crack, or cross-section that prove that there are really cracking after indenting. Please provide at least high magnification of the cracks. In my opinion, the lines in 6b and 6f can not be related to the microcracks. If so, these cracks are quite strange. Figure 6 is also very difficult to understand.

Response: Actually, the γ-TiAl intermetallc compound is a brittle material. Moreover, after oxidation at high temperature, the specimen was covered by the oxides scale, which means that the cracks can be introduced by the indenter. The microcracks is a common phenomenon for a brittle material tested by indenter. Here, these prepared specimens were put into the quartz tube to ensure the loss of oxide flakes for 100 h at 1000℃, and weighted the total mass of quartz tube every 10 h by an analytical balance. The oxides scale formed on the surface of γ-TiAl alloy. The Rockwell hardness tester (HRC, 150 kg) was utilized to introduce the artificial microcracks onto the surface of specimens treated by NaF aqueous solution after isothermal oxidation for 100 h at 1000℃. The brittle oxides scale can be easily destroyed with microcracks.

  1. To confirm the effect of fluorine treated sample, the same SEM, TEM analysis should be presented on the non-treated sample. Without this control information, one can not draw a conclusive statement on the effect of treated samples.

Response: Actually, γ-TiAl alloys has developed at least 50 years. The information of non-treated sample including results tested by XRD, SEM, TEM have been reported extensively, such as references [1], [15], [18] and [19]. Moreover, the fluorine treated γ-TiAl alloys exhibit a strong oxidation resistance, which has been confirmed by M. Fröhlich, R. Braun, and C. Leyens et al, please see the reference [11], [13], [14], [23] and [27]. The aim of this manuscript is to cater to demand of industrial applications for low-cost, the smart reintroduction of fluorine in the fluorine treated γ-TiAl alloys system for engineering repairs, particularly in repairing of the surface protective oxide film of blade and turbine disk.

Reviewer 2 Report

Dear Editor and authors

Does the author present well the results in the title "Fluorine effect for improving oxidation resistance of Ti-45Al-8.5Nb alloy at 1000 0C", still i see there is lacking in novelty, new concept in this work.  There is no scientific soundness in fluorine treated oxidized sample of Ti-Al-Nb sample, first and second step. 

I hereby reject the article based on scientific soundness. 

Author Response

  1. Does the author present well the results in the title "Fluorine effect for improving oxidation resistance of Ti-45Al-8.5Nb alloy at 1000℃", still i see there is lacking in novelty, new concept in this work. There is no scientific soundness in fluorine treated oxidized sample of Ti-Al-Nb sample, first and second step.

Response: Actually, γ-TiAl alloys has developed at least 50 years. The information of non-treated sample including results tested by XRD, SEM, TEM have been reported extensively, such as references [1], [15], [18] and [19]. Moreover, the fluorine treated γ-TiAl alloys exhibit a strong oxidation resistance, which has been confirmed by references [11], [13], [14], [23] and [27]. The aim of this manuscript is to cater to demand of industrial applications for low-cost, the smart reintroduction of fluorine in the fluorine treated γ-TiAl alloys system for engineering repairs, particularly in repairing of the surface protective oxide film of blade and turbine disk. For the current γ-TiAl alloy, a nominal composition of Ti-45Al-8.5Nb (at.%) was fabricated by a vacuum induction melting furnace. This alloy is getting closer to applications at high temperature (above 900℃) in recent years, although the other γ-TiAl alloys have been applied almost 10 years (such as 4822 alloys at 750℃). The origin-idea of this manuscript to treatment—oxidation—cracked specimens—treatment again—oxidation again for the Ti-45Al-8.5Nb alloy was inspired by the engineering repairing of the Ni-based superalloys. Therefore, we think the innovations in this research field of the fluorine treated γ-TiAl alloys has been improved step by step, and the current results is helpful for the promoting application of the Ti-45Al-8.5Nb alloy.

Reviewer 3 Report

  1. «This section may be divided by subheadings. It should…» What's this?
  2. «A casting γ-TiAl alloy ingot with a nominal composition of Ti-45Al-8.5Nb (at.%) was 60 fabricated by a vacuum induction melting furnace.» Need more information or a link to a paper where this alloy was researched.
  3. Figure 2. It is necessary to indicate the available phases on the image.
  4. «SEM image of as-cast TiAl…» And in the work was used some other alloy, besides cast?
  5. From figure 3, it is not clear what the red and black graphs demonstrate. This requires an explanation in the figure.
  6. «It indicated the preferential growth of these phases, which related to the improved oxidation resistance…» If the authors assert about the growth of phases, it is necessary to present the results of quantitative X-ray diffraction analysis.
  7. The conclusion is more like an Abstract and needs to be changed.

Author Response

Reviewer #3 (Comments to the Author):

  1. This section may be divided by subheadings. It should What's this?

Response: Good suggestion! We had modified this part as marked in manuscript.

  1. A casting γ-TiAl alloy ingot with a nominal composition of Ti-45Al-8.5Nb (at.%) was 60 fabricated by a vacuum induction melting furnace. Need more information or a link to a paper where this alloy was researched.

Response: Good suggestion! Actually, γ-TiAl alloys has developed at least 50 years. The information of non-treated sample including results tested by XRD, SEM, TEM have been reported extensively, such as references [1], [15], [18] and [19]. The current alloy in this manuscript has cast by a vacuum induction melting furnace, which is same as the fabrication of reverences [15], [18] and [19].

  1. Figure 2. It is necessary to indicate the available phases on the image.

Response: Good suggestion! We have redrawn the figure. 2 as follows:

  1. SEM image of as-cast TiAl. And in the work was used some other alloy, besides cast?

Response: No, the TiAl alloy was cast for preparing specimens in this article. We had employed “TiAl alloy” instead of the “as-cast TiAl alloy”.

  1. From figure 3, it is not clear what the red and black graphs demonstrate. This requires an explanation in the figure.

Response: Good suggestion! We have redrawn the figure. 2 as follows:

  1. It indicated the preferential growth of these phases, which related to the improved oxidation resistance. If the authors assert about the growth of phases, it is necessary to present the results of quantitative X-ray diffraction analysis.

Response: Good suggestion! We had deleted the sentence “It indicated the preferential growth of these phases, which related to the improved oxidation resistance.”. The direct evidence of the improved oxidation resistance for NaF solution treated γ-TiAl specimens was shown in Fig. 3.

  1. The conclusion is more like an Abstract and needs to be changed.

Response: Good suggestion! We had modified the conclusion as follows: A simple and effective strategy containing a two-step spraying and oxidation processes have been conducted to produce Ti-45Al-8.5Nb alloy with dense Al2O3 film for high temperature application, particularly in repairing of the surface protective oxide film of blade and turbine disk. The experimental process follows the fluorine treated TiAl alloy—isothermal oxidation—cracked specimen—fluorine treatment again—isothermal oxidation again. The oxidation resistance of the specimens were studied by isothermal oxidation at 1000℃, and found that has been significantly ameliorated due to the formation of a protective Al2O3 film under the help of fluorine. For the oxidized specimen with the artificial defect, the microcracks act as a rapid-diffusion passage for Ti, Al and O elements. The more activity and diffusion rate of Ti than that of Al lead to a quick growth of TiO2 toward outside in a short time. But the fine Al2O3 constituted the continuous film at the bottom of the oxide scale after isothermal oxidation. In particular, Al2O3 particles grew toward substrate and embodied in the γ-TiAl and α2-Ti3Al phases, leading to the good oxidation adhesion.

Round 2

Reviewer 1 Report

Dear Authors, #

thank you for answering my questions, although, most of them is said to be  a future work. Please however, do include the answer into your revised manuscript as much as possible. What I asked could be possibly asked by the readers themselve when they read your articles. That is why your answers need to be in the revised text, not only for me to know. 

Other comments are: 

  • Please add clearly the statement on the wetting behaviours of NaF on the surface of pristine sample, as well as on the oxidized and precracked sample.
  • Please clarify how Ti and Al react with NaF to form TiF3 and AlF3? Is this the chemical reaction? or just absorption process? If it is a reaction, how deep is the reaction zone? Is this time dependce process or not? Even there is an information in literature on the reactions, this should be investigated for your sample since it is your idea with spraying methode.

Author Response

Reviewer #1 (Comments to the Author):
1. thank you for answering my questions, although, most of them is said to be a future work. Please however, do include the answer into your revised manuscript as much as possible. What I asked could be possibly asked by the readers themselves when they read your articles. That is why your answers need to be in the revised text, not only for me to know.

Other comments are:

Please add clearly the statement on the wetting behaviours of NaF on the surface of pristine sample, as well as on the oxidized and precracked sample.

Response: With respect to your good suggestions, we have carefully revised our manuscript, incorporated and highlighted in the new version resubmitted. In order to complete the state of the art of the paper, we have increased the number of references during the revision processes. The authors appreciated the course of revision which was beneficial and pleasant, with thanks to you.

In addition, the wetting behaviours of NaF on the surface of pristine sample, as well as on the oxidized and precracked sample is discussed in detail as follow:

Since the wettability of the NaF-containing aqueous solution drops on γ-TiAl alloy had an important influence on the oxidation process, contact angle tests were carried out to evaluate the effect of the NaF solution drops on their wetting performance on sample’s surfaces, as shown in Figure. 3. The initial surface of pristine Ti-45Al-8.5Nb alloys sample was hydrophilic, where the wetting angle was 39.6° (Figure. 3a). The results represent the good wettability of interface between NaF-containing aqueous solutions and Ti-45Al-8.5Nb alloys. In addition, the effects of the wetting angle of interface between NaF-containing aqueous solutions and oxidized TiAl alloys with and without spraying NaF-containing aqueous solution are investigated in Figure 3b and 3c. The results shown that the contact angle increased after 1000℃/100 h isotherm oxidized treatment. Also, from the comparison of the Figure 3b and 3c, we can find that fluorine-treated is benefit for improving wettability of the oxidation Ti-45Al-8.5Nb alloys surface.

Figure 3. The wettability of the NaF-containing aqueous solution on the γ-TiAl alloys: (a) pristine γ-TiAl alloy, (b) Fluorine-treated γ-TiAl alloy after isothermal oxidation, (c) bare-TiAl alloy after isothermal oxidation.

Reviewer 2 Report

Author has slightly improve this article. However i am not totally convinced about the scientific soundness, which is lacking mostly. Author could support using mechanism of repair the oxidized surface by NaF treatment. Some chemical diffusion and treatment at the surface. 

One related article may need to follow

Fabrication of thermal plasma sprayed NiTi coatings possessing functional properties

. Coatings 11 (5), 610,2021

Author Response

  1. Author has slightly improve this article. However i am not totally convinced about the scientific soundness, which is lacking mostly. Author could support using mechanism of repair the oxidized surface by NaF treatment. Some chemical diffusion and treatment at the surface.

One related article may need to follow

Fabrication of thermal plasma sprayed NiTi coatings possessing functional properties. Coatings 11 (5), 610,2021

Response: With respect to your good suggestions, we have carefully revised our manuscript, incorporated and highlighted in the new version resubmitted. In order to complete the state of the art of the paper, we have added the description of mechanism during the revision processes, the in detail as follow:

Remarkably, similar results were reported by previous work [31, 32, 33, 39], indicating that the Ti-fluoride and Al-fluoride could generate because of the chemical reactions between the fluorine and metals (including Ti and Al) by fluorine treatment (such as ion implantation, anodic oxidation and spray). After oxidation for a short time, the exclusive Al2O3 and minor TiO2 form on the surface of specimens due to the volatilisation of Ti-fluoride. With a longer oxidation period, the fluorine promotes the formation of a continuous and dense Al2O3 layer, which is consist with chemical elements distribution results in Figure. 5 and Figure. 7. The oxidation approaches of relevant reactions follow the Me (Me=Ti, Al)→MeF (fluorides)→MeOF (oxy-fluorides)→MeO (oxides); or/and Me→MeF→MeO [34]. The continuous Al2O3 layer retards the inner diffusion of oxygen and outside diffusion of metals. Under the Al2O3 layer, the fluorine-aided selective transport of Al by the Al-fluoride is related with the partial pressures of p(AlF) and p(F2) [31, 40], such as the minimum p(F) of Ti0.5Al0.5 was 1.30×10-8 bar at 1000℃. If the balance was broken or peeled off at local region of the specimen caused by the failure of oxides scale during oxidation [26], the Ti and Al will diffuse competitively from the broken region, and followed by an accelerated failure of oxides scale. The current experimental process follows the fluorine treated TiAl alloy-isothermal oxidation-cracked specimen-fluorine treatment again-isothermal oxidation again. For the current second-treated oxidized-TiAl specimens, the cracked-surface contacted firstly by the F solution. The Al monofluoride on the surface of specimens has the highest vapor pressure than other fluorides [41, 42]. Because of the vapor vocalization of the Al monofluoride, the remained-Ti element on the surface of specimens formed TiO2 during oxidation again at high temperature, leading to the visible strips of TiO2-rich particles. Under the oxides scale, the partial pressure of oxygen decreased steeply. The Al monofluoride reacted with the oxygen to form Al2O3 at high temperature (900℃) with the lowest formation energy [32, 33, 41]. The artificial microcracks act as a rapid-diffusion passage for Ti, Al and O elements. The more activity and diffusion rate of Ti than that of Al lead to a quick growth of TiO2 toward outside in a short time. But the fine Al2O3 constituted the continuous film at the bottom of the oxide scale after isothermal oxidation. In particular, Al2O3 particles rather than TiO2 particles grew toward substrate and embodied in the γ-TiAl and α2-Ti3Al phases, leading to the good adhesion. Therefore, the γ-TiAl alloys treated by fluorine has the excellent anti-oxidation property at high temperature.

In addition, the related article followed as References 15:

Fabrication of thermal plasma sprayed NiTi coatings possessing functional properties. Coatings 11 (5), 610,2021:

The authors appreciated the course of revision which was beneficial and pleasant, with thanks to you.

Reviewer 3 Report

The changes provided are not enough, and unfortunately the manuscript should be rejected. Even though the authors have changed the conclusion, it still looks like an abstract or conclusion to a technical report. There are no specific values that justify the chosen method. Because of this, I believe that the relevance of the work is not sufficiently substantiated, which requires major changes.

Author Response

  1. The changes provided are not enough, and unfortunately the manuscript should be rejected. Even though the authors have changed the conclusion, it still looks like an abstract or conclusion to a technical report. There are no specific values that justify the chosen method. Because of this, I believe that the relevance of the work is not sufficiently substantiated, which requires major changes.

Response: With respect to your good suggestions, we have carefully revised our manuscript, incorporated and highlighted in the new version resubmitted. In order to complete the state of the art of the paper, we have rewritten the conclusion during the revision processes. The authors appreciated the course of revision which was beneficial and pleasant, with thanks to you.

Round 3

Reviewer 1 Report

Dear Authors, 

thank you to add wetting analysis! I think the quality of the manuscript is improved a lot. 

From my previous report, the second comment: 

  • Please clarify how Ti and Al react with NaF to form TiF3 and AlF3? Is this the chemical reaction? or just absorption process? If it is a reaction, how deep is the reaction zone? Is this time dependce process or not? Even there is an information in literature on the reactions, this should be investigated for your sample since it is your idea with spraying methode.

was not answered or discussed. 

If possible please consider these comments on final version before publication. 

Best regards and good luck with the manuscript. 

Author Response

Response: These are really good questions to improving this manuscript. The previous work of XPS results suggest that the chemical reaction will happen between Ti and F, also between Al and F (The reference is “Oxidation of Metals (2018) 90: 617”). And the XPS results are as follows:

The related reactions between M and F (M =Ti and Al) are as following:

  • M+0.5F2 → MF(g)
  • MF(g)+0.5O2 → MOF(g)
  • MOF(g)+0.5O2 → MaOb +F2
  • MF(g)+O2 → MaOb +F2

According to the Friedle’s results (Oxidation of Metals (2018) 89: 123), the F-implanted sample before oxidation at high temperature has a thickness of several micrometers tested by TOF-SIMS depth profiles. So, we can guess that the depth of these chemical reactions of current results is not over the F-implanted sample’s depth of the reference. However, according the current results, we cannot obtain whether they depend on the time.

Please find the detailed response attached.

Reviewer 2 Report

The article has improved with each step of revision. I would recommend author to come up with mechanism of effect of Fluorine on oxidation resistance of Ti-45Al-8.5Nb alloy at 1000 ℃. The mechanism and the reaction process is highly necessary to understand the process how it works. 

It is important.

Author Response

Response: The chemical reaction process between M and F (M =Ti and Al) are as following:

  • M+0.5F2 → MF(g)
  • MF(g)+0.5O2 → MOF(g)
  • MOF(g)+0.5O2 → MaOb +F2
  • MF(g)+O2 → MaOb +F2

The mechanism of the current results has tried to provide, as follows:

The oxidation approaches of relevant reactions follow the Me (Me=Ti, Al)→MeF (fluorides)→MeOF (oxy-fluorides)→MeO (oxides); or/and Me→MeF→MeO [34]. The continuous Al2O3 layer retards the inner diffusion of oxygen and outside diffusion of metals. Under the Al2O3 layer, the fluorine-aided selective transport of Al by the Al-fluoride is related with the partial pressures of p(AlF) and p(F2) [31, 40], such as the minimum p(F) of Ti0.5Al0.5 was 1.30×10-8 bar at 1000℃. If the balance was broken or peeled off at local region of the specimen caused by the failure of oxides scale during oxidation [26], the Ti and Al will diffuse competitively from the broken region, and followed by an accelerated failure of oxides scale. The current experimental process follows the fluorine treated TiAl alloy-isothermal oxidation-cracked specimen-fluorine treatment again-isothermal oxidation again. For the current second-treated oxidized-TiAl specimens, the cracked-surface contacted firstly by the F solution. The Al monofluoride on the surface of specimens has the highest vapor pressure than other fluorides [41, 42]. Because of the vapor vocalization of the Al monofluoride, the remained-Ti element on the surface of specimens formed TiO2 during oxidation again at high temperature, leading to the visible strips of TiO2-rich particles. Under the oxides scale, the partial pressure of oxygen decreased steeply. The Al monofluoride reacted with the oxygen to form Al2O3 at high temperature (900℃) with the lowest formation energy [32, 33, 41]. The artificial microcracks act as a rapid-diffusion passage for Ti, Al and O elements. The more activity and diffusion rate of Ti than that of Al lead to a quick growth of TiO2 toward outside in a short time. But the fine Al2O3 constituted the continuous film at the bottom of the oxide scale after isothermal oxidation. In particular, Al2O3 particles rather than TiO2 particles grew toward substrate and embodied in the γ-TiAl and α2-Ti3Al phases, leading to the good adhesion. Therefore, the γ-TiAl alloys treated by fluorine has the excellent anti-oxidation property at high temperature.

Please find the detailed response attached.

Reviewer 3 Report

The authors have made the necessary changes to the text in accordance with their responses to previous comments. I would like more precise values in the conclusion:
1. Compared to what did the oxidation resistance improve and by how much?
2. How long does it take for a film to form and what can be considered "fast" formation?
3. How much does adhesion increase?

All this should be clarified.

Author Response

Response: As suggested, we have further rewritten the conclusion in the revised manuscript, and added more precise values in the conclusion.

A simple and effective strategy involving a two-step spraying and oxidation processes have been carried out to produce Ti-45Al-8.5Nb alloys with a dense Al2O3 film for the application at a high temperature. The main results are shown as follows:

  • The oxidation resistance was significantly improved due to the formation of a protective Al2O3 film induced by the fluorine effect during isothermal oxidation at 1000℃ (6.03 mg/cm2 weight gain for the bare Ti-45Al-8.5Nb alloys, but 0.81 mg/cm2 for the fluorine-treated Ti-45Al-8.5Nb alloys).
  • The Ti diffused along the artificial microcracks, resulting in a quick growth of TiO2 in a short time (2 h), while a continuous Al2O3 film generated underneath the oxide scale during the further isothermal oxidation (100 h).
  • The Al2O3 particles between the alloy and continuous Al2O3 film were reflected in the γ-TiAl and α2-Ti3Al phases, which improved the adhesion of oxides scale (fluorine-treated specimen has a deep yellow oxide scale on the surface without any exfoliation). These findings represented a significant progress towards reducing the high temperature oxidation resistance of the TiAl intermetallic by surface fluorine modification.
